# The Impact of the Vanadium Oxide Addition on the Physicochemical Performance Stability and Intercalation of Lithium Ions of the TiO_2_-rGO-electrode in Lithium Ion Batteries

**DOI:** 10.3390/ma13041018

**Published:** 2020-02-24

**Authors:** Beata Kurc, Marcin Wysokowski, Łukasz Rymaniak, Piotr Lijewski, Adam Piasecki, Paweł Fuć

**Affiliations:** 1Institute of Chemistry and Electrochemistry, Faculty of Chemical Technology, Poznan University of Technology, Berdychowo 4, PL-60965 Poznan, Poland; 2Institute of Chemical Technology and Engineering, Faculty of Chemical Technology, Poznan University of Technology, Berdychowo 4, PL-60965 Poznan, Poland; marcin.wysokowski@put.poznan.pl; 3Institute of Combustion Engines and Transport, Faculty of Transport Engineering, Poznan University of Technology, Piotrowo 3, PL-60965 Poznan, Poland; lukasz.rymaniak@put.poznan.pl (Ł.R.); piotr.lijewski@put.poznan.pl (P.L.); pawel.fuc@put.poznan.pl (P.F.); 4Faculty of Mechanical Engineering and Management, Institute of Materials Science and Engineering, Poznan University of Technology, Jana Pawla II 24, PL-60965 Poznan, Poland; adam.piasecki@put.poznan.pl

**Keywords:** solvothermal synthesis, reduced graphene oxide, anatase, vanadium oxide, lithium ion batteries

## Abstract

This work determines the effect of the addition of various amounts of vanadium oxide on the work of a cell built from a hybrid V_x_O_y_-TiO_2_-rGO system in a lithium-ion cell. Moreover, a new method based on solvothermal chemistry is proposed for the creation of a new type of composite material combining reduced graphene, vanadium oxide and crystalline anatase. The satisfactory electrochemical properties of VxOy-TiO_2_-rGO hybrids can be attributed to the perfect matching of the morphology and structure of VxOy-TiO_2_ and rGO. In addition, it is also responsible for the partial transfer of electrons from rGO to VxOy-TiO_2_, which increases the synergistic interaction of the VxOy-TiO_2_-rGO hybrid to the reversible storage of lithium. In addition a full cell was created LiFePO_4_/V_x_O_y_-TiO_2_-rGO. The cell showed good cyclability while providing a capacity of 120 mAh g^−1^.

## 1. Introduction

Continuous technological progress means that people are constantly looking for new solutions that will enable them to work more efficiently and make the most of their potential. The constant movement and very active lifestyle of millions of people from around the world forces somehow being in constant motion, which makes them portable devices. Lithium-ion cells are widely used in modern technologies as separate systems.

The automotive industry continues working to introduce and promote energy saving and environmentally friendly solutions. The use of alternative fuels, hybrid and electric drives, as well as fuel cells, could be included among those [1,2,3,4]. This requires introducing significant changes in vehicle construction and even infrastructure adaptation, e.g., building battery charging stations.

Electric and hybrid drives used in city buses are characterized by many favorable operational parameters in urban operating conditions and it is in this vehicle group that ongoing efforts are being undertaken to introduce various types of innovations to improve the efficiency of electrical solutions, mainly the energy storage systems. Their addition to hybrid systems makes it possible to turn off the internal combustion engine during stops and to use only the electric drive, e.g., in zero emission zones [5]. In this type of systems, energy is recovered when the vehicle brakes, converting the kinetic energy of the vehicle into electricity.

The search for suitable cathodic and anodic materials is still underway. Among the former there are, among others, vanadium oxides. Due to the preservation of good properties during loading and unloading processes, relatively easy transfer of lithium ions, several degrees of oxidation (V^3+^, V^4+^, V^5+^), ease of synthesis and lack of toxicity, they are a good basis for working with lithium-ion cells. They also provide good cycling stability [6,7,8,9]. Vanadium oxides are regularly used in research works around the world. New synthetic methods are still being developed that use this compound in the most diverse fields of science. During the preparation of the research material, various compilations using vanadium oxide are used. Such materials can be prepared by various methods including hydrothermal synthesis, precipitation [10] and sol gel synthesis [11]. In one of the recent works of Ceder et al. [12] the slow diffusion kinetics in V_2_O_5_ electrodes was also indicated, particularly in the stable α-V_2_O_5_ phase associated with the conversion during the discharge process of fully demagnesiated α-V_2_O_5_ into a coexisting two-phase structure with fully magnesiated α-V_2_O_5_ and fully demagnesiated α-V_2_O_5_ phases. The relatively large theoretical capacity when intercalating two lithium Li^+^ ions per unit cell is the main advantage of this oxide.

On top of this the addition of porous V_2_O_5_ was shown to provide interconnected network for efficient transport of Li^+^ across the entire structure. Moreover, the interconnected network infuses higher mechanical robustness in order to cope with the active materials volume expansion while also retaining dimensional stability at the electrode level. The feasibility of this approach has been successfully demonstrated, but the V_2_O_5_ electrodes can from the material’s poor structural stability, low-diffusion coefficient of lithium ions (~ 10^−12^ cm^2^ s^−1^) and moderate electronic conductivity (10^−2^ to 10^−3^ S cm^−1^) when used in rechargeable Li-ion batteries.

The unique structure could provide numerous electrolyte access channels to facilitate rapid diffusion of lithium ions into the electrode material. It also affected the short semiconductor diffusion for lithium due to the thin layer of VxOy. It also influenced the high electricity conductivity of the entire electrode based on the graphene network and the highest content of electrochemically active material in the electrode. The hybrid obtained using this method can meet all kinetic requirements for efficient charging and discharging of the ideal electrode material-i.e., rapid ion diffusion. As a result, the graphene structure and VxOy enable the electrode to work efficiently: charging and discharging with a long life cycle while maintaining a highly reversible capacity. The addition of vanadium oxide reduces structural changes during work, prevents, among others material swelling.

The following table shows sample combinations of reagents used in vanadium oxide methods as an electrode material. Depending on the purpose of the test material being prepared, different compilations of used substances are selected. The review of these substances aims to show how many possibilities exist for the use of vanadium oxide in a wide variety of lithium-ion cell research. The most common combinations with its use: rGO-V_2_O_5_; TiO_2_-V_2_O_5_.

rGO is widely used as a template for solvothermal deposition of V_2_O_5_ [13] as well as TiO_2_ [14,15]. Surprisingly, there are no reports to date concerning the use of rGO as a template for the development of vanadia–titania nanocomposites for Li-ion batteries.

The unique structure could provide numerous electrolyte access channels to facilitate rapid diffusion of lithium ions into the electrode material. It also affected the short semiconductor diffusion for lithium due to the thin layer of VxOy. It also influenced the high electricity conductivity of the entire electrode based on the graphene network and the highest content of electrochemically active material in the electrode. The hybrid obtained using this method can meet all kinetic requirements for efficient charging and discharging of the ideal electrode material-i.e., rapid ion diffusion. As a result, the graphene structure and VxOy enable the electrode to work efficiently: charging and discharging with a long life cycle while maintaining a highly reversible capacity. The addition of vanadium oxide reduces structural changes during work, prevents, among others material swelling.

Therefore, we decided to develop, for the first time, a solvothermal reaction for the synthesis of novel V_x_O_y_-TiO_2_-rGO nanocomposites and to investigate their potential application in the construction of a new generation of Li-ion batteries. Incorporation of rGO into V_x_O_y_-TiO_2_ is beneficial in terms of the cell stability. The cell does not lose significant capacity during extended work.

## 2. Materials and Methods

### 2.1. Synthesis of V_x_O_y_-TiO_2_-rGO Oxide Nanocomposites

For the typical synthesis of V_x_O_y_-TiO_2_-rGO oxide nanocomposites, a solution consisting of 20 mL of H_2_O and 40 mL of CH_3_COOH (Sigma-Aldrich, Hamburg, Germany) underwent vigorous stirring at room temperature; then 2 mL of rGO (Merck, Hamburg, Germany), 10 mL of titanium(IV) butoxide (Sigma-Aldrich), an appropriate quantity of NH_4_VO_3_ (Merck) and 1 mL of NH_4_OH (Merck) were added gradually. The detailed quantities of reagents used for the synthesis of selected samples are given in Table 1, Figure 1.

### 2.2. Electrode and Electrolyte

The tested anodes were analyzed: V_x_O_y_-TiO_2_-rGO on a copper foil (Hohsen, city, Japan)|LiPF_6_ solid salt in liquid EC/DMC (1:1)|Li. The ratio of components was V_x_O_y_-TiO_2_-rGO (75%), AB (15%), PVdF (10%), (by weight). In order to get rid of the NMP solvent, the electrodes were dried under vacuum for 24 h at 120 °C.

### 2.3. Procedures and Measurements

The morphology and microstructure of the prepared samples were examined on the basis of the SEM images recorded from an EVO40 scanning electron microscope (Zeiss, Oberkochen, Germany). The particle size distributions were determined using a Zetasizer Nano ZS (Malvern Instruments Ltd.

A Vertex 70 spectrophotometer (Bruker, Bremen, Germany) was used to record the Fourier transform infrared spectra (FTIR). Analysis was conducted on the materials in the form of tablets, with the WAXS method used for crystal phase identification analysis. The results were analyzed using X-RAYAN software. To determine the electrochemical properties of the cells electrochemical impedance spectroscopy (EIS) and galvanostatic charging/discharging tests were used. The cycling measurements were taken using a Gammry Industries (Louis Drive Warminster, PA 18974) multichannel electrochemical system at different current density values, while the multichannel potentiostat (Gammry) was used for cyclic voltammetry (*CV*).

In order to assess the morphologies of particles, the scanning electron microscope (SEM), model MIRA3, produced by Tescan (Brno–Kohoutovice, Czech Republic) was used. An accelerating voltage of 20 kV was applied. The chemical compositions of powders were measured by ULTIM MAX 65 microanalyzer with EDS method (Oxford Instruments) at the same accelerating voltage i.e., 20 kV. The contents of elements such as Ti, O and V were analyzed. To reduce samples charging, the graphite film with thickness of about 20 nm was coated on powders using a JEE 4B vacuum evaporator (JEOL, Dearborn Road Peabody, MA, USA).

The electrodes: Li: ca. 45 mg (0.785 cm^2^), V_x_O_y_-TiO_2_-rGO: 2.5–3.5 mg, were separated by a glass microfiber GF/A separator (Whatmann, 0.4–0.6 mm thick). The electrodes were placed in an adapted Swagelok^®^ connecting tube. Cells were assembled in a glove box in a dry argon atmosphere.

## 3. Results

### 3.1. Dispersive and Structural Properties of the Obtained Materials

The SEM images presented below indicate that TiO_2_-rGO (S_2_V_0_Ti_1_) obtained by a solvothermal method without the presence of vanadium precursor has a uniform spherical shape with particle sizes ranging between 50 nm and 150 nm, and a tendency to form aggregates. This observation was confirmed by measurements of particle size distribution (Figure 2).

The graph obtained confirms a bimodal particle size distribution resulting from the aggregation of nanoparticles into sub-micrometric aggregates. The SEM images and the particle size distribution graphs presented for samples obtained with increasing quantities of vanadium precursor show an increasing tendency towards the formation of large conglomerates composed of spherical nanoparticles (Figure 2).

To confirm composite formation and to identify the functional groups present on the surface of the obtained materials, FTIR spectroscopy was applied. Figure 3a shows the spectra recorded for V_x_O_y_-TiO_2_-rGO composite samples and for TiO_2_-rGO as a reference sample. The fundamental vibrations in all of the TiO_2_-based composites appear in the FTIR spectra as a very intense broad band, ascribed to stretching vibrations of Ti–O bonds (550–653 cm^−1^) [16,17]. The next band is visible at 1103 cm^−1^, and according to the literature [18] is related to stretching vibrations of unshared V=O bonds [19]. The highest intensity of this band is observed for the sample obtained with the highest quantity of vanadium precursor. The band visible at 1333 cm^−1^ is a result of stretching vibrations of C=O and C–OH bonds. The multiple bands visible in a range from 1414 cm^−1^ to 1551 cm^−1^ are related to the graphitic core of reduced graphene oxide [20,21]. The intense band at ~ 1630 cm^−1^ corresponds to deformation vibrations of adsorbed water molecules [16], but also overlaps with a band related to C=C sp^2^ bonded carbon atoms [22,23]. The intense wide band at 3407 cm^−1^, with a shoulder at 3241 cm^−1^, is attributed to stretching vibrations of O–H groups [16].

The XRD patterns of the samples are presented in Figure 3b. All patterns confirm that the materials synthesized via the solvothermal method exhibit very high crystallinity. All observed 2Θ peaks are characteristic for anatase (JCPDS, No. 21–1272). It can be observed that the incorporation of vanadium precursor does not lead to the formation of any new crystalline phase, and does not influence the crystallinity of the obtained materials.

Nevertheless, the fact that no vanadium phases (such as the V_2_O_5_ orthorhombic phase) are observed in these XRD measurements indicates that the vanadium oxide is deposited as an amorphous coating on the TiO_2_ nanoparticles.

In Figure 4 the results of chemical composition studies were presented as the EDS spectrums. In the tables, the concentrations of the studied elements (Ti, O and V) as well as the standard deviation were presented. The studies from the area of 1 mm^2^ were carried out.

The S_2_V_0_Ti_1_ sample was characterized by the highest content of titanium and oxygen, whereas for the S_2_V_1_Ti_1_ sample the highest content of vanadium, i.e., 4.5 wt% was measured.

In Figure 5 SEM image and EDS concentration maps of V, O and Ti for the sample S_2_V_1_Ti_1_ were presented. The maps are shown using a 12 color scale. The colors corresponded to the concentration of a selected element, the white pixels meant the highest concentration of element and the black areas corresponded to a concentration equal to zero. The numerical values corresponded to the number of counts. After applying individual areas on each other, it could be stated that areas with light and dark colors overlapped. Thus, the resulting material in the form of a powder with particle sizes from 0.4 to 5 µm was homogeneous in respect of chemical composition. The intensity was compatible to the results presented in the form of EDS spectrums in Figure 4.

### 3.2. Electrochemical Properties

Nanoscience and nanotechnology offers some new perspectives. Novel techniques and increased understanding of material science have made it possible to create and craft appropriate anode materials for new LIBs. Thanks to the appropriate physicochemical characteristics of the structures, lithium storage has been achieved. In addition, a high lithium-ion flux at the electrode/electrolyte interface has been obtained, the diffusion distance for both Li and electron ions has been reduced, and anode volume changes during the charging/discharging process have been significantly reduced. By combining all of the aforementioned features into new devices it should be possible to create new high energy density and high power density devices.

The V_x_O_y_-TiO_2_-rGO composite anode exhibited stable cycle performance, while retaining a discharge capacity of 160 mAh g^−1^ after the 50th cycle. This is comparable to the capacities reported in the literature [24,25,26,27,28,29,30,31,32,33,34,35,36] (see Table 2).

The results of the cyclability study of all investigated materials are presented in Figure 6. The cycling behavior of the V_x_O_y_-TiO_2_-rGO electrode is plotted at a low current density of 50 mA g^−1^, as shown in Figure 5. S_2_V_0_Ti_1_ exhibit the initial discharge–charge capacities of 110 and 120 mAh g^−1^, respectively. In the first cycle, a relatively high specific capacity was observed, which is a common phenomenon. In the literature, this is translated as the formation of solid electrolyte interphase (SEI).

When cycling to the 30th cycle, the capacity is 100 mAh g^−1^ and then it gradually increases to 85 mAh g^−1^ in the 500 mA g^−1^ (Table 3).

Moreover Table 3 shows the multiple-current galvanostatic result of the V_x_O_y_-TiO_2_-rGO electrode, with discharge capacities of: S_2_V_0_Ti_1_:120–85 mAh g^−1^; S_2_V_0.25_Ti_1_: 134–95 mAh g^−1^; S_2_V_0.5_Ti_1_: 155–118 mAh g^−1^ at current densities of 50, 100, 150, 200, 300 and 500 mA g^−1^, respectively.

Figure 6 shows charge discharge capacities depending on the number of cycles, and the profile of charging curves relative to the potential. As you can see, the most stable and simultaneously reversible arrangement turned out to be S_2_V_1_Ti_1_. Coulombic efficiency is close to 96% and slight differences in charging curves at 1st, 2nd and 30th cycles.

The electrode material is a composite and therefore the capacities given have not been corrected for the vanadium oxide content. It was observed that, after 120 cycles (S_2_V_0.5_Ti_1_) at higher current density, the prepared material still shows an impressive discharge capacity of ~ 118 mAh g^−1^ at a current density of 500 mA g^−1^ (Table 3). All samples show relatively good capacity retention after cycles with gradually increasing currents. One may notice that S_2_V_1_Ti_1_ shows reversible capacities of about 182 mAh g^−1^ at 50 mA g^−1^ (Figure 6), and this value drops to 120 mAh g^−1^.

The specific capacity is low-suggesting that lithium incorporation is limited under these conditions and that surface vanadate oxides do not significantly improve ion diffusion at the electrolyte interface. To determine the performance of the S_2_V_1_Ti_1_ porous electrode in practical applications, a complete cell with a commercial LFP cathode was assembled (Figure 7).

Charge/Discharge capacity of the commercial LiFePO_4_ half cell at C/5 is shown in Figure 7a. Figure 6 shows the dependence of Li/S_2_V_1_T_1_ cell capacity on the cycle, taking into account the current densities at which the measurements were made. As can be seen in the chart above, the capacity decreases with subsequent cycles, especially at high current density values. Comparing the above results with literature data we get very similar values. In studies on the impact of the V_x_O_y_ structure in lithium-ion cells, the results obtained at 50 cycles and the current of the same density oscillate in the range of 200–150 mAh g^−1^ depending on the sample. However, in studies using reduced graphene oxide and V_x_O_y_ in lithium-ion cells, values after 50 cycles were approximately 200 mAh g^−1^.

The full cell provides a discharge capacity of 158 mAh g^−1^ in the first cycle (Figure 7c), which decreases slowly as the number of cycles increases. At a discharge capacity of 137 mAh g^−1^ after 30 cycles. Previous studies have shown that a decrease in capacity in a full cell can be the result of many parameters, among others: different current density between two electrodes, loss of Li ions in the direction of SEI formation and other side reactions or the use of inappropriate electrolyte.

It was found that the system achieves quite good efficiency, with a return current density of up to 50 A g^−1^, because the discharge capacity reaches initial values (Figure 7a). When analyzing the SEI layers, it is important to remember about good synergy with the electrode material, in addition, it is important that it does not dissolve in the electrolyte (especially at high temperatures). It should be a good electronic insulator and an efficient ion conductor for lithium ions, while removing the solvation coating around lithium ions. The latter aims to avoid co-intercalation of the solvent, which is associated with the exfoliation of active material [19].

The SEI layer is formed on the electrode surface due to electrolyte decomposition (solvent and salt) at potential below 1 V vs. Li /Li^+^ (usually 0.7 V vs. Li/Li^+^ for most electrolytes). The potential for SEI formation, composition and stability depends on many factors, such as electrolyte formulation, charge and discharge rate, etc. [19].

Power losses are referred to as “irreversible”. However, it is very well known that in many cases the original capabilities are “recoverable”. It has been proven that the loss of capacity results from many factors that cause the increase in SEI-this is the case with carbon anodes. The “irreversible” loss of capacity in a balanced cell is caused by the phenomenon in which the lithium involved in the reaction comes from the cathode. Similar phenomena are known when SEI is formed on the cathode and the lithium does not have enough lithium to restore the cathode to full capacity after charging [20].

The nature of the electrolyte and the composition of the solvent components have a significant impact on the nature of SEI. This is a key issue that governs the performance of both hard and soft carbon anode materials.

The prepared composite can, therefore, endure significant changes, including low or high current densities, and yet retain stability upon cycling. This is an advantage for LIBs with high power and long lifecycle as it can increase their abuse tolerance.

The storage capacity that occurs through an intercalation path, is closely associated to the surface area, morphology, crystallinity and its orientation. Soft carbons are commonly used and well accepted in the battery industry.

The presented Scheme 1 emphasizes that both the current collectors and the electrolyte side come from electrons and lithium ions in the electrodes. It was found that there are a number of resistances that are characteristic of a composite electrode, including surface resistances of so-called active materials, which are associated with conduction paths.

Ion resistance in electrolyte and solid state diffusion in active materials is directly related to ionic conductivity. For larger-size lithium-ion batteries, the distribution of reactions can occur within composite electrodes. This is most likely due to the fact that the electrode reaction occurs preferentially in areas with lower resistance. When creating composite electrodes, remember the relationship between electron/ion conductivity and the distribution of reactions in electrodes and battery performance. The distribution model has previously been studied from a theoretical point of view. There are several reports in which the distribution of composite electrode reactions is directly observed. The reactions between the electrode and the electrolyte depend on: material composition, its morphology, porosity of the composite material used for electrode construction. In addition, the morphological image determines their effective ionic and electronic conductivity.

The tuning of such composite electrodes, however, has been mostly based on the intuition and experience of the person performing it. This is because it is especially difficult to distinguish the electronic conductivity and the ionic conductivity in composite electrodes [34,35,36,37].

In addition, heterogeneous aggregation of nanostructures may affect the performance of individual cells. For longer nanostructures, the intercalation of the second lithium ion is limited, and the phase δ completely transforms into the γ phase without irreversible reactions with slower kinetics, which in turn increases the structural stability of the material. In the literature, it can be argued that the electrochemical properties of nanostructures strongly depend on the particle size: the smaller the particle size, the lower the polarization and higher cell capacity.

To increase the conductivity and specific surface area of active materials, graphene and nanomaterials are used. This also causes a significant reduction in the phenomenon of agglomeration during cyclical work.

In numerous literature we already find systems containing vanadium oxides and graphene nanocomposites, such as VxOy @ G. The problem, however, is the controlled synthesis of vanadium oxides on graphene nanoparticles of various structures. This is an important aspect from the point of view of electrochemical properties obtained [38,39,40,41,42,43]. It is important to obtain a stable, homogeneous and composite with good structural reserve, based on V_2_O_5_/reduced graphene oxide.

The synergistic effect between vanadium oxide and the graphene substrate affects performance and excellent cyclic stability. This is influenced by: the growth of V_2_O_5_ nanoparticles on reduced graphene oxide, which also ensures good electronic conductivity of electrode materials. Porous space created by interconnected nanocomponents can improve the availability of electrolyte with electrode materials. What’s more, the thickness of one thin component can reduce the diffusion of Li^+^ ions. It also limits the transport of electrons; and the porous structure may better account for changes in volume during the cycle [44,45]. Figure 8 shows representative cyclic voltammograms (CVs) of S_2_V_0_Ti_1_, S_2_V_0.25_Ti_1_, S_2_V_0.5_Ti_1_ and S_2_V_1_Ti_1_ for the first cycles at a scan rate of 0.5 mV s^−1^ in a voltage window of 0–3 V.

Cyclic voltammetry is a technique used to determine the range of potentials at which the cell will exhibit the best properties. It also provides us with information on the processes taking place on the electrodes. Figure 6 shows the cycle of the tested cell with the feed rate of 0.5 mV s^−1^. The reduction process in cells using vanadium oxide as the electrode material is 1.6 V. The oxidation reaction corresponds to a potential of 2.25 V. In similar studies regarding modified V_2_O_5_ used in lithium-ion cells [46], the reduction peak appeared in the vicinity of 2.3 V. In contrast, the peak corresponding to the oxidation reactions occurred in the 2.5 V range, which is close to the above results.

As can be seen, the reduction peak shifts to the low potential region with increasing content of TiO_2_, whereas the oxidation peaks shift to the high potential region, suggesting polarization under a high scan rate. The cyclic voltammograms (CVs) is widely performed to study the oxidation/reduction process and Li intercalation/deintercalation behavior in the electrode reactions.

Poor electrochemical performance can often still be caused by a minimal increase in volume as well as irreversible reactions during cyclic operation. This is also affected by the inadequate internal conductivity of V_2_O_5_ [45,46,47]. Various solutions can be found in the literature: intervention in structure design or planned morphology, for example V_2_O_5_ with a carbon coating. In the literature, we see that simple and direct strategies to solve this problem are used—the introduction of conductive carbon into the composite, physical mitigation of volume growth by creating an empty structure or exposing certain aspects by regulating morphology. Further network level design is relatively small, especially on an atomic scale, but Cao et al. [48] found that amorphous vanadium pentoxide exhibits better energy density and cycle stability than crystalline ones when used in batteries. Good properties have been shown to result from the isotropic percolated diffusion network in amorphous V_2_O_5_, which facilitates fast-faradic reactions. It has also been noticed that crystalline V_2_O_5_can provide higher capacity than amorphous V_2_O_5_ at high current densities, while amorphous provides higher capacity at low current densities [15,49].

Figure 9 shows scanning electron microscopy (SEM) images of pristine electrodes and after electrochemical cycling. The S_2_V_0_Ti_1_ and S_2_V_1_Ti_1_ particles after electrochemical cycling are covered with a film and small aggregates. This ‘micro-roughness’ may indicate the formation of an SEI layer.

Robust intercalation tunnels and the ability of many electrons to pass, produces a V_2_O_5_ electrode with high power efficiency, which requires the use of dimensional parameters and transport to overcome the complex environment of physicochemical reactions. Likely by the directions of ion diffusion towards the electrode and load transfer to the electrode/electrolyte interface, which created a new aspect of research that scientists are currently trying to solve.

Using the thermodynamic expression, it should be noted that the available energy depends on the electrode materials, it is difficult to fully release the theoretical value of the chemical energy accumulated in V_2_O_5_ due to the limitations of ion diffusion kinetics and V_2_O_5_ computer conductivity.

## 4. Conclusions

The solvothermal method has been shown to be an efficient tool for the preparation of V_x_O_y_-TiO_2_-rGO materials with uniform spherical morphology. The addition of vanadium precursor to the reaction system facilitates the aggregation of particles into large agglomerates. XRD measurements indicate that the vanadium atoms are well incorporated in the TiO_2_ crystalline structure. The work has demonstrated that V_x_O_y_-TiO_2_-rGO displays improved electrochemical stability upon the reported lithiation and delithiation, which effectively improves the long-term electrochemical performance and maintains the specific capacity well. The V_x_O_y_-TiO_2_-rGO microparticles synthesized as described here may be a promising candidate as an anode material for future application in LIBs.

It has been proved that a relatively porous structure allowed the use of the surface area to quickly replenish electrolyte ions. It has also been found that the surface porosity of various structures can create an interconnected network for better Li^+^ transport throughout the structure. In addition, this limited the significant increase in volume during electrode operation.

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
