# Peer review of "The Impact of the Vanadium Oxide Addition on the Physicochemical Performance Stability and Intercalation of Lithium Ions of the TiO2-rGO-electrode in Lithium Ion Batteries"

_materials, 2020, doi:10.3390/ma13041018_

Round 1

Reviewer 1 Report

Kurc et al. reported the preparation of VxOy-TiO2-rGO nanocomposites by a solvothermal method, and their application for lithium-ion batteries, which exhibits satisfactory electrochemical properties with a capacity of 120 mAh g-1 and good stability. The addition of vanadium precursor suppresses the aggregation of particles and facilitates them into large conglomerates. However, to meet the publishable criterion of the journal, a minor revision is still required.

If the author wants to highlight the impact of VxOy addition on lithium storage, the author should directly summarize the impact by using a few words and emphasize it in the title. It would be better if it gets down to very fundamental physics. At present, I cannot see the emphasis of this work from the title. The author should first explain why they choose to add vanadium oxide on TiO2-rGO electrode in the introduction part. What is the contribution of vanadium oxide for lithium storage? Is there any synergistic effect with TiO2-rGO? The author highlights the controllable preparation of VxOy-TiO2-rGO and their superior lithium storage. It would be better if the author further explains the connection between the heterostructures and lithium storage in detail.

Author Response

Thank you very much for your comments and thorough review! I hope that the answers are enough to publish our work.

Reviewer 1

Kurc et al. reported the preparation of VxOy-TiO2-rGO nanocomposites by a solvothermal method, and their application for lithium-ion batteries, which exhibits satisfactory electrochemical properties with a capacity of 120 mAh g-1 and good stability. The addition of vanadium precursor suppresses the aggregation of particles and facilitates them into large conglomerates. However, to meet the publishable criterion of the journal, a minor revision is still required.

Answer: Thank you for your time, kind revision, and valuable remarks for upgrading the manuscript.

If the author wants to highlight the impact of VxOy addition on lithium storage, the author should directly summarize the impact by using a few words and emphasize it in the title. It would be better if it gets down to very fundamental physics. At present, I cannot see the emphasis of this work from the title.

Answer: The title has been changed:

The impact of the vanadium oxide addition on the stability of physicochemical performance and intercalation of lithium ions of the TiO2-rGO - electrode in lithium ion batteries

The author should first explain why they choose to add vanadium oxide on TiO2-rGO electrode in the introduction part.

Answer: It has been addend:

The unique structure could provide numerous electrolyte access channels to facilitate rapid diffusion of lithium ions into the electrode material. It also affected the short semiconductor diffusion for lithium due to the thin layer of VxOy. It also influenced the high electricity conductivity of the entire electrode based on the graphene network and the highest content of electrochemically active material in the electrode. The hybrid obtained using this method can meet all kinetic requirements for efficient charging and discharging of the ideal electrode material - i.e. rapid ion diffusion. As a result, the graphene structure and VxOy enable the electrode to work efficiently: charging and discharging with a long life cycle while maintaining a highly reversible capacity. The addition of vanadium oxide reduces structural changes during work, prevents, among others material swelling.

What is the contribution of vanadium oxide for lithium storage? Is there any synergistic effect with TiO2-rGO? The author highlights the controllable preparation of VxOy-TiO2-rGO and their superior lithium storage. It would be better if the author further explains the connection between the heterostructures and lithium storage in detail.

Answer: The addition of graphene and nanomaterials can not only increase the conductivity and specific surface area of active materials, but also prevent their agglomeration during the cycle. In recent years, vanadium oxides and graphene nanocomposites such as VxOy@G, have been preparated. However, controlled production of vanadium oxides on graphene nanoparticles of various structures is rarely reported, which has a large impact on electrochemical properties [1-6].

The vanadium precursor can increase homogeneously on the reduced graphene oxides and the resulted composites can be changed into V2O5/reduced graphene oxide composites with good structural reservation.

The excellent rate performance and superior cyclic stability can be attributed to the synergistic effects between vanadium oxide and graphene substrates, which include the following aspects: the growth of V2O5 nanosheets on reduced graphene oxide can ensure the good electronic conductivity of the electrode materials. In the literature we can read that the porous space created by the interconnected large nanosheets can improve the accessibility of the electrolyte with the electrode materials. Moreover the thin nanosheet thickness can greatly reduce the Li+ ions diffusion and electron transportation distances; and the porous structure may better accommodate the volume changes upon cycling [7].

  1. Yoo E, Kim J, Hosono E, Zhou H-S, Kudo T, Honma I. Large reversible Li storage of graphene nanosheet families for use in rechargeable lithium ion batteries. Nano lett. 2008;8:2277–82.
  2. Wang X, Shi G. Flexible graphene devices related to energy conversion and storage. Energy Environ Sci. 2015;8:790–823.
  3. Gwon H, Kim H-S, Lee KU, Seo D-H, Park YC, Lee Y-S, Ahn BT, Kang K. Flexible energy storage devices based on graphene paper. Energy Environ Sci. 2011;4:1277–83.
  4. Yang S, Gong Y, Liu Z, Zhan L, Hashim DP, Ma L, Vajtai R, Ajayan PM. Bottom-up approach toward single-crystalline VO2-graphene ribbons as cathodes for ultrafast lithium storage. Nano Lett. 2013;13:1596–601
  5. Zhang Y, Pan A, Liang S, Chen T, Tang Y, Tan X. Reduced graphene oxide modified V2O3 with enhanced performance for lithium-ion battery. Mater Lett. 2014;137:174–7.
  6. Xu J, Li Z, Zhang X, Huang S, Jiang S, Zhu Q, Sun H, Zakharova GS. Self-assembled V3O7/graphene oxide nanocomposites as cathode material for lithium-ion batteries. Int J Nanotech. 2014;11:808–18
  7. Y. Liu, Y. Wang, Y. Zhang, S. Liang, A. PanControllable Preparation of V2O5/Graphene Nanocomposites as Cathode Materials for Lithium-Ion Batteries, Nanoscale Res Lett. 2016; 11: 549-553.

Reviewer 2 Report

In the reviewed paper, Beata Kurc et al. study the impact of the addition of vanadium during the synthesis of mixed TiO2-rGO materials.

A lot of work and characterization have been done however, I do not completely agree with the data interpretation

The authors reported that vanadium is incorporated into TiO2 (based on XRD in the conclusion). I would expected no vanadium incorporation based on the XRD because there is no XRD peak shifts in the series that would mean (no change in the structure).
Moreover, the authors attributed detected a vanadyl bond (V=O) by I spectroscopy which does not fit with the fact that vanadium is incorporated into the TiO2 anatase. There would be only V-O bonds and not V=O. Did the authors think about the possibility of a vanadium based amorphous coating? The experimental part should be more detailed: how the powder is recovered after the synthesis …filtration or centrifugation …..

From my expertise once titanium butoxide is introduced in aqueous solution like here, white precipitate (TiO2) can directly be observed, is it the case here? In this case, a vanadium coating would be more expected.
Is the solution yellow at the end of the synthesis?

It is strange that vanadium is detected by EDS even for S2VOTi1 where no vanadium was used for the synthesis The ref 9 deals with the optical properties of ZnO, I do not understand the link with this work and the ref 24 does not deal with Zr doped V2O5 as citd in the manuscript. I think all the bibliography should be double-checked. The manuscript shoud be proofread again there are several small mistakes (L63, L210 (2 times 50mA/g)…)

Author Response

Thank you very much for your comments and thorough review! I hope that the answers are enough to publish our work.

Reviewer 2

In the reviewed paper, Beata Kurc et al. study the impact of the addition of vanadium during the synthesis of mixed TiO2-rGO materials.

A lot of work and characterization have been done however, I do not completely agree with the data interpretation

The authors reported that vanadium is incorporated into TiO2 (based on XRD in the conclusion). I would expected no vanadium incorporation based on the XRD because there is no XRD peak shifts in the series that would mean (no change in the structure).
Moreover, the authors attributed detected a vanadyl bond (V=O) by I spectroscopy which does not fit with the fact that vanadium is incorporated into the TiO2 anatase. There would be only V-O bonds and not V=O. Did the authors think about the possibility of a vanadium based amorphous coating?

Answer: Thank you for your valuable remark. Indeed vanadium oxide can be deposited as amorphous coating therefore we changed the text.

The experimental part should be more detailed: how the powder is recovered after the synthesis …filtration or centrifugation …..

From my expertise once titanium butoxide is introduced in aqueous solution like here, white precipitate (TiO2) can directly be observed, is it the case here? In this case, a vanadium coating would be more expected.
Is the solution yellow at the end of the synthesis?

Answer: Thank you for your valuable remark. Indeed, solution is pale yellow after synthesis. We agree that vanadium oxide is deposited as a coating on the TiO2 surface.

It is strange that vanadium is detected by EDS even for S2VOTi1 where no vanadium was used for the synthesis

Answer: Thank you for your valuable remark. We repeated the measurements, seems that it was kind of external impurity, in present version the figure is corrected

Round 2

Reviewer 2 Report

Thank you for you reply.

The authors answered to the comment / questions and improved the manuscript.

The manuscript can be accepted in its present form.